# SarQoL Questionnaire in Community-Dwelling Older Adults under EWGSOP2 Sarcopenia Diagnosis Algorithm: A New Screening Method?

**DOI:** 10.3390/ijerph19148473

**Published:** 2022-07-11

**Authors:** Carlos Guillamón-Escudero, Angela Diago-Galmés, David Zuazua Rico, Alba Maestro-González, Jose M. Tenías-Burillo, Jose M. Soriano, Julio J. Fernández-Garrido

**Affiliations:** 1Hospital General Universitari de Castelló, 12004 Castelló de la Plana, Spain; carlos_ge@hotmail.es; 2Hospital Universitario de La Plana, 12540 Villarreal, Spain; angela94dg@gmail.com; 3Department of Medicine, Nursing Area, University of Oviedo, 33006 Oviedo, Spain; zuazuadavid@uniovi.es; 4Hospital Universitario Central de Asturias, 33011 Oviedo, Spain; 5Department of Preventive Medicine, Hospital Pare Jofré, 46017 Valencia, Spain; tenias_jma@gva.es; 6Food & Health Lab, Institute of Materials Science, University of Valencia, 46980 Valencia, Spain; jose.soriano@uv.es; 7Joint Research Unit on Endocrinology, Nutrition and Clinical Dietetics, University of Valencia-Health Research Institute La Fe, 46026 Valencia, Spain; 8Department of Nursing, Faculty of Nursing and Podiatry, University of Valencia, 46001 Valencia, Spain; julio.fernandez@uv.es

**Keywords:** sarcopenia, older adults, quality of life, SarQoL, older people, muscle strength

## Abstract

This article is an observational and cross-sectional study that related the result obtained in the questionnaire for the evaluation of quality of life related to muscle mass (SarQoL) and the prevalence of sarcopenic pathology measured under the EWGSOP2 algorithm. Participants were 202 community-dwelling older adults living in Valencia, Spain. The prevalence of sarcopenia in men was 28.9%, while in women it was 26.2%. In the case of the SarQoL questionnaire, the mean score obtained for men was 75.5 and 72.6 for women, showing significant differences in both sexes between the results obtained by the group with and without sarcopenia. After the exhaustive data analysis, a high discriminative capacity for sarcopenic disease was found in the SarQoL questionnaire total score and in domains 2 (locomotion), 4 (functionality) and 5 (activities of daily living). In accordance with the existing controversy regarding the use of SARC-F as a screening method for sarcopenia, the authors pointed out the capacity of domain 2 (locomotion) in isolation as a possible screening method for this disease, exposing a high risk of suffering sarcopenia when scores in this domain were below 60 points. Further research is needed to develop new lines of research as these showed in this work, as well as new and easily applicable screening methods for sarcopenia in clinical practice, that allow a rapid detection of this disease in the community.

## 1. Introduction

Sarcopenia, according to the diagnostic criteria of the European Working Group on Sarcopenia in Older People (EWGSOP2) [1], is defined as a generalized and progressive skeletal muscle disorder that is associated with a higher probability of adverse outcomes including falls, fractures, physical disability and mortality. In this latest revision of sarcopenia’s definition, special attention is paid to muscle strength, which has been shown as a key factor in the prediction of disease-related adverse effects. In this latest revision of the disease definition, particular attention is paid to muscle strength, which has been shown to be the key factor in predicting disease-related adverse effects [2,3,4,5]. Sarcopenic pathology is related to age, its appearance being more common after the sixth decade of life [6,7]; despite this, with the implementation of better diagnostic resources, the increasing knowledge and value of the disease among the different health specialists, it is becoming increasingly common to find a greater number of cases in a younger population [8].

Traditionally, sarcopenia’s presence has been linked to health institutions [9,10,11,12], since most of the publications focused on institutionalized patients, who were more accessible to researchers and on whom it could be appreciated to a greater extent the effects of sarcopenia. Nowadays, some researchers stress the need to study sarcopenia in community-dwelling older adults [13,14,15,16,17], since, traditionally, the pathology in this population group was found to be underdiagnosed and may constitute a relevant point of action in terms of public health policies and prevention of higher sarcopenic stages, which through its complications could increase the risk of hospitalization notably [18,19].

At the same time, the analysis of perceived quality of life proved to be important for government and health institutions, since it allows health-related reforms planning according to the specific needs of the population [20].

Complications derived from sarcopenia are related, mainly, with mobility deterioration from patients as well as with a frailty increase, malnutrition, a progressive loss of autonomy and, globally, a decrease in perceived quality of life [20,21,22,23,24]. With the aim of evaluating quality of life and its relationship with muscle mass, in 2016, the SarQoL questionnaire [25] (Sarcopenia Quality of Life) was developed, which was validated for the Spanish population in 2020 [26,27].

This study aim was to evaluate possible differences in terms of quality of life quantified by the SarQoL questionnaire [27] in community-dwelling men and women depending on the presence of sarcopenic disease. Secondarily, researchers started from the hypothesis of the existence of a possible discriminatory capacity of sarcopenic pathology through the SarQoL total score [27] or any of its domains, and whether it can be used as an additional screening tool along with the SARC-F questionnaire [28] proposed by EWGSOP2 [1].

## 2. Material and Methods

### 2.1. Study Design and Recruitment of the Sample

Participants of this observational, descriptive and cross-sectional study were recruited in a center for the elderly in Benimaclet (Valencia), managed by Valencia City Council. This center collaborated with the University of Valencia together with the city council in the framework of the commitment acquired regarding the Chair of Healthy, Active and Participative Aging.

The study was sampled by convenience as no randomization method was used. The final sample obtained was of 202 community-dwelling adults over 65 years old; 38 (18.8%) of them were men and 164 were women (81.2%). The inclusion criteria applied to the participants for participation in the research were the following: being over 65 years of age, being enrolled in the center for the elderly where the study was conducted and having completed the informed consent and all phases of the study. The exclusion criteria were the following: presenting diseases that implied a severe deterioration of muscle mass (Parkinson’s, Alzheimer’s, severe cognitive impairment, stroke, muscular dystrophy and cancer) and/or being absent from the center on days on which the study was performed.

Participation in this study was voluntary; all participants completed the relevant informed consent for participation. This research met the standards established by the Declaration of Helsinki [29] and was approved by the bioethics committee of the University of Valencia (Spain) (n/1139186).

### 2.2. General Assessments

To study general sociodemographic variables of the participants, researchers used a specific ad-hoc questionnaire generated for the occasion. The variables studied were: sex, coexistence at home, age, blood pressure record, heart rate record and chronic diseases (hypertension, diabetes, osteoporosis and dyslipidemia).

#### 2.2.1. Sarcopenia

Sarcopenia diagnosis was made using the latest EWGSOP2 standards [1]. As a result, the sample was divided into two groups: with sarcopenia and without sarcopenia. Participants in the sarcopenic group must have had probable, confirmed or severe sarcopenia, while the group without sarcopenia comprised the rest of the participants.

##### Handgrip Strength (Upper Body Strength) and Sit-to-Stand Test (Lower Body Strength)

To determine muscle strength, a diagnostic test targeting the two main body segments was performed. In the upper segment, a Jamar 5030J1 manual dynamometer, with a measurement scale of 0–90 kg/f and with an accuracy of ±2 kg, was used to determine the grip strength of the participants [1,30,31,32]. In the lower segment, the sit-to-stand test was performed [1,33] to assess the strength in the subjects’ legs. The cut-off points referred by the EWGSOP2 [1] were used for the categorization of the results.

Participants who showed low muscle strength in the upper body segment and/or low muscle strength in the lower segment were categorized as probable sarcopenia, with their amount of muscle mass evaluated in the next step of the algorithm presented by the EWGSOP2 [1].

##### Appendicular Skeletal Muscle Mass (ASMM)

Appendicular muscle mass was evaluated using the formula proposed by Kyle et al. [34] to obtain necessary data to complete the equation; the electrical bioimpedance measurement was used with a TANITA DC430MA-S scale (Tokyo, Japan) with a precision of 0.05 kg. In order to obtain accurate measurements, researchers implemented the latest recommendations for performing this technique [35].

According to the EWGSOP2 consensus [1], subjects who presented low muscle strength and also obtained ASMM values lower than 15 kg (muscle mass deficit) were considered sarcopenic confirmed cases; their physical performance was evaluated to determine the severity of the sarcopenic pathology.

##### Physical Performance (Gait Speed)

Physical performance of the participants was evaluated using the gait speed test [1,36]. This test consisted of measuring the time taken by the subjects to walk a distance of 4 m at a constant and habitual speed. According to the EWGSOP2 consensus [1], participants who presented low muscle strength, low muscle mass and values lower than 0.8 m/s (low physical performance) were categorized as subjects with severe sarcopenia, this being the maximum expression of the disease’s development.

#### 2.2.2. Physical Activity Quantification

Researchers used the International Physical Activity Questionnaire Adapted for the Elderly (IPAQ-E) [37,38] with the study participants with the aim of evaluating the amount of daily mean metabolic equivalents (METS) that subjects were able to develop.

#### 2.2.3. Frailty Status

The frailty of the subjects was evaluated using the Frailty Instrument for Primary Care of the Survey of Health, Aging and Retirement in Europe (SHARE-FI) [39]. This tool is validated for the Spanish population in the age range of the study participants [40].

The SHARE-FI instrument [40] consists of 5 sections that evaluate fatigue, appetite, manual grip muscle strength, functional difficulties and the frequency of physical activity performed; each item offers a numerical result that is processed through the calculator offered by the tool. Finally, it gives a categorization of the subject in three possible states: frail, pre-frail and non-frail.

#### 2.2.4. Nutritional Status

The nutritional status of the participants was evaluated using the Self-MNA test [41] validated for populations older than 65 years [41,42]. This test was self-administered according to its original format; despite this, its completion was supervised by researchers in order to resolve any doubts that might arise among the participants.

The Self-MNA test [41] categorized participants into three states based on the final score obtained: normal nutritional status (12–14 points), risk of malnutrition (8–11 points) and malnutrition (0–7 points).

### 2.3. Quality of Life (SarQoL)

To assess participants’ quality of life, the Spanish translated version of the SarQoL questionnaire [27] was used, offered by developers. All the questions asked in this questionnaire, except 7, 14 and 22, use a Likert scale of frequency or intensity; in turn, these questions are categorized into seven domains that assess different dimensions related to quality of life: physical and mental health (D1), locomotion (D2), body composition (D3), functionality (D4), activities of daily living (D5), leisure activities (D6) and fears (D7). Each domain, as well as the total score of the questionnaire, is quantified on a scale from 0 to 100, in which a higher score implies a better quality of life.

The administration of this questionnaire was carried out by the researchers, who clarified and/or resolved the doubts that arose during its completion.

To obtain the corresponding detailed results of the questionnaires once completed with the participants’ information, researchers used the official platform offered by the creators (www.SarQoL.org, accessed on 25 June 2022). This questionnaire has strong internal consistency, as can be consulted in various publications [27,43,44,45].

### 2.4. Statistical Analysis

Statistical analysis was carried out using the IBM SPSS Statistics v. 24 software for Windows (IBM Corp., Armouk, NY, USA). The normality of the data was evaluated using the Shapiro–Wilk’s test. Descriptive statistics were calculated for sociodemographic variables (percentages or mean ± SD), results of the SarQoL questionnaire [27] and its domains (mean ± SD) and for the different variables related to physical performance, as well as the diagnosis of the sarcopenia (mean ± SD). The differences between groups with and without sarcopenic disease were determined using the Student’s *t* test and Mann–Whitney U test, establishing significant differences with *p* values ≤ 0.05 in all tests. The Spearman correlation model was used to evaluate the relationships between the different variables that make up the diagnosis of sarcopenia and the results obtained in the SarQoL questionnaire [27] and in its domains. Variables that showed significant differences were evaluated in a later step using a multiple linear regression model in order to observe their influence on the score obtained for each of the domains and on the total score of the SarQoL questionnaire [27].

Additionally, a ROC curve analysis model was developed where the ability to discriminate sarcopenia as a pathology through the total score of the SarQoL test [27] and the three most influential domains in the general result of the questionnaire was evaluated. Subsequently, knowing the specificity and sensitivity thresholds in the sarcopenia diagnosis of each score obtained in the total SarQoL questionnaire [27], researchers designed possible risk areas that would allow the use of this test as a diagnostic-preventive factor.

## 3. Results

The study sample was composed of a total of 202 subjects; the mean age was 73 ± 7 years. The youngest participant was a 65-year-old woman, and the oldest was an 85-year-old woman. Of the total sample, 7.4% had confirmed or severe sarcopenic pathology and 26.7% had some of the disease stages. The mean value obtained regarding participants’ total quality of life was 75.3 ± 10.1 points.

Table 1 summarizes general characteristics of the sample distributed by sex based on sarcopenic disease presence. In the men group, age showed significant differences in terms of the presence of sarcopenia; men who presented the disease were older. Regarding frailty, men and women who presented a pre-frail or frail state were more prevalent in the sarcopenic disease group (*p* = 0.015 and *p* = 0.001, respectively). In the nutritional aspect, significant differences (*p* = 0.017) were only found in the men group, where the results obtained showed a higher prevalence of malnutrition in the group of sarcopenic participants. No significant differences were found in terms of the prevalence of sarcopenia for BMI or chronic diseases under study in either sex. The quality of life of the participants analyzed through the SarQoL questionnaire [27] yielded significantly lower results in the sarcopenia group in both men and women, the values obtained in most of the domains of the questionnaire being significantly lower in the groups who presented the disease. Finally, the parameters related to physical activity studied showed significantly lower results in the sarcopenia groups regardless of sex, with the exception of the total amount of ASMM.

Relationships between main variables related to sarcopenia and the SarQoL questionnaire domains [27], as well as the total result obtained in the questionnaire, were analyzed according to sex and can be consulted in detail in Table 2.

Thus, we can observe how all the variables, except the total amount of ASMM, showed significant differences regarding the correlations for the final result of the SarQoL questionnaire [27] regardless of sex. All variables related to sarcopenia diagnosis presented worse results, as seen with the lower score obtained in terms of quality of life. Domains three, six and seven were ones that showed the least significant correlations regardless of gender in terms of their result and their relationship with main variables related to sarcopenia diagnosis.

Table 3 shows the results of the multiple linear regression analysis for all of the SarQoL questionnaire domains and its total score. Most of the variables related to sarcopenia diagnosis presented worse results given the lower scores in the domains or in the total score of the SarQoL questionnaire [27]. This finding associated worse results in these variables with a worse score in terms of quality of life for the participants.

After what was observed in Table 2 and Table 3 and taking into account the high relationship between most of the variables related to sarcopenia and scores obtained in different domains and in the total SarQoL questionnaire score [27], researchers decided to perform an analysis of ROC curves in order to assess the discriminatory capacity of the questionnaire and its domains for the diagnosis of sarcopenia, regardless of sex; this model is shown in Figure 1 and can be consulted in detail in Table 4. Areas under the curve were greater than 0.7 for domains two, four and five and for the total SarQoL [27] score, this latter being the one that showed a greater discriminatory capacity for the diagnosis of sarcopenic disease (area under the curve: 0.756). Additionally, the authors performed the same test excluding patients with probable sarcopenia from the population with the disease (it can be seen in depth in Table 5).

Taking into account the sensitivity and specificity in the sarcopenia diagnosis for each of the total scores of the questionnaire obtained, researchers delimited three risk areas in terms of the probability of suffering sarcopenia, regardless of sex, based on results obtained in the SarQoL questionnaire [27] test; this model can be consulted in Figure 2. Thus, obtaining a result of fewer than 60 points in the questionnaire would imply a high risk of suffering from sarcopenia; a result between 61 and 85 implies a moderate risk; and a score higher than 86 implies a low risk of suffering from sarcopenia.

## 4. Discussion

The results of this study showed a significant relationship between quality of life quantified through the SarQoL test [27] and the presence of sarcopenia in the study subjects. As far as the authors know, this is the first article in which the SarQoL questionnaire [27] was used to quantify community-dwelling older people’s quality of life with or without sarcopenia. Furthermore, this work presents a pioneering approach in terms of evaluating the ability of the questionnaire to discriminate sarcopenic disease regardless of the participant’s sex in the total SarQoL score and its domains.

The updated EWGSOP2 algorithm [1] was used for the diagnosis of sarcopenia despite the existence of other diagnostic tools in the literature [46,47,48,49]. This algorithm is widely endorsed by the scientific community [11,14,50,51,52,53], being a reference in the great part of the world.

Our study used a sample size and yielded results similar to those obtained by Geerinck et al. [54] in their initial research to observe the feasibility of the SarQoL questionnaire as a screening method for sarcopenia. The cut-off score obtained in their study was 52.4 points when assessing the risk between low and high of suffering from the disease, while the researchers of this article placed it at 60 points. This difference could be due to the fact that the entire sample of the work presented by Geerinck et al. [54] was not collected outside health institutions or because no cases with probable sarcopenia were included in its intervention group.

Additionally, the area under the curve obtained by Geerinck et al. [54] for the total score of the SarQoL questionnaire was higher (0.771) than the score obtained by the researchers of this study when the same conditions were applied to the group with the disease (Table 5) or including the cases of probable sarcopenia (0.756).

Our work presented a prevalence of confirmed and severe sarcopenia similar to that found by Kim et al. [50] in community-dwelling older people using the algorithm proposed by EWGSOP2 [1]. Furthermore, consistent with Volpato et al. [55] and Cruz-Jentoft et al. [56], a significant relationship was found between age and the presence of sarcopenic pathology in men; this same relationship could not be demonstrated in women, unlike what was stated by other authors [57,58] in their studies. Significant differences were also found regarding the level of education and the prevalence of sarcopenia in women, with sarcopenia being more prevalent in women with primary education or without studies; this situation is similar to that expressed by Dorosty et al. [59] for both sexes in their publication, despite being an institutionalized population and having evaluated sarcopenic pathology following the EWGSOP recommendations [60] published in 2010. The significant increase in pre-frail and frail status in sarcopenic groups regardless of sex was found in our study, as in that presented by other authors [23,24,61]. Regarding chronic diseases studied, no significant differences were found when they were related to sarcopenia regardless of sex, although other authors [58,62] reported an osteoporosis increase in sarcopenic individuals.

Regarding quality of life, significant differences were found in the total score obtained in the SarQoL questionnaire [27], observing a worse quality of life in individuals with sarcopenia regardless of gender; this fact is also contemplated in publications made by other authors [20,45,63], despite not using the same tool for quantifying quality of life. Most of the SarQoL domains [27] presented worse scores in the group with sarcopenia, except in domain three (body composition) and seven (fears) in both groups, and domain six (leisure activities) also in the female group. These domains did not seem to have much influence on the quality of life of subjects with sarcopenic pathology. More studies are needed to broaden the discussion regarding this last statement.

The main variables for sarcopenia diagnosis showed significant differences when they were related to the total SarQoL score [27], with worse results in these variables obtained with participants with a lower quality of life. This situation was not manifested in the case of the ASMM, since it did not show significant differences when relating it to the score obtained in the questionnaire. The absence of a relationship between the amount of ASMM and quality of life seems to be in agreement with that expressed by other authors [2,3,4,5] regarding the need to reduce the relevance of this variable in the diagnosis of sarcopenia; in fact, the EWGSOP in its latest update [1] already introduced the importance of strength, physical performance or the quality of muscle mass against the detriment of the amount of ASMM as a diagnostic variable.

After consulting some publications in which the use of the SARC-F test [28] recommended by the EWGSOP2 [1] as a screening tool for sarcopenia was discussed, the authors detected some uncertainty regarding the current capacity of the test to detect early possible cases of sarcopenia [17,54,64]. That is why, after observing the significant differences in terms of quality of life between subjects living in the community with or without sarcopenic pathology, regardless of sex, and taking into account the purpose of this possible new adaptation of the SarQoL questionnaire, as well as the original function of the SARC-F as a screening questionnaire, the authors decided that it was relevant to include the subjects with probable sarcopenia within the group with sarcopenia in the study, since the early detection of a presarcopenic situation is also relevant in current medicine. The results obtained show that the total score of the questionnaire and domains two (locomotion), four (functionality) and five (activities of daily living), autonomously, would allow for detecting cases of sarcopenia. The development of risk intervals for sarcopenia detection was carried out based on the total SarQoL score [27], as it was the analyzed variable of the questionnaire that presented a greater area under the curve (0.756).

The SarQoL questionnaire [27] has an approximate duration of 10 min according to its developers. We consider that the total duration of the questionnaire can be an inconvenience for its use as a screening tool; for this reason, we suggest the use of domain 2 (locomotion) autonomously. This domain presented an area under the curve of 0.749, is composed by only 9 items and requires less time to complete. The authors suggest carrying out the SarQoL questionnaire in nursing or medicine primary care consultations with the vulnerable population over 65 years of age as a screening method at least once a year. If the value obtained is fewer than 85 points, researchers recommend performing the complete battery of tests for the diagnosis of sarcopenia explained in the EWGSOP2 [1] algorithm.

This study has severe limitations: the sampling carried out was non-random despite the voluntary nature of the participants; it was not multicenter; the small sample could lead the authors to bias in the results or some false-negative cases. Despite this, it also shows strengths: the application of the updated EWGSOP2 algorithm [1] in its entirety, the pioneering vision with which the discrimination capacity of the questionnaire for the diagnosis/prevention of sarcopenia is evaluated, as well as the use of a control group that allows comparing the differences between the sarcopenic and non-sarcopenic groups in community-dwelling older adults.

More research is needed to continue exploring: the ability of the SarQoL questionnaire [27] to detect sarcopenic disease early, its potential as a screening tool together with the SARC-F test [28] proposed by the EWGSOP2 [1] and the importance of the quality of life within sarcopenic disease, responding directly to the subjective assessment of patients regarding the limitations caused by this disease. Quality of life must be a direct feedback channel for the production of primary and secondary health policies that are as close as possible to the citizen and therefore to the future patient.

## 5. Conclusions

In summary, this study shows a significant reduction in the quality of life of sarcopenic community-dwelling subjects regardless of sex when evaluated using the EWGSOP2 diagnostic algorithm [1] and SarQoL questionnaire [27]. Participants presented significantly lower values for the main variables related with sarcopenia diagnosis when the results obtained in the analysis of quality of life were worse, except ASMM. The SarQoL [27], in its entirety, or using domains 2, 4 and 5 autonomously, demonstrated discriminatory capacity for sarcopenic pathology diagnosed under the EWGSOP2 algorithm [1], as well as a potential use as an element of population screening.

## Figures and Tables

**Figure 1 ijerph-19-08473-f001:**
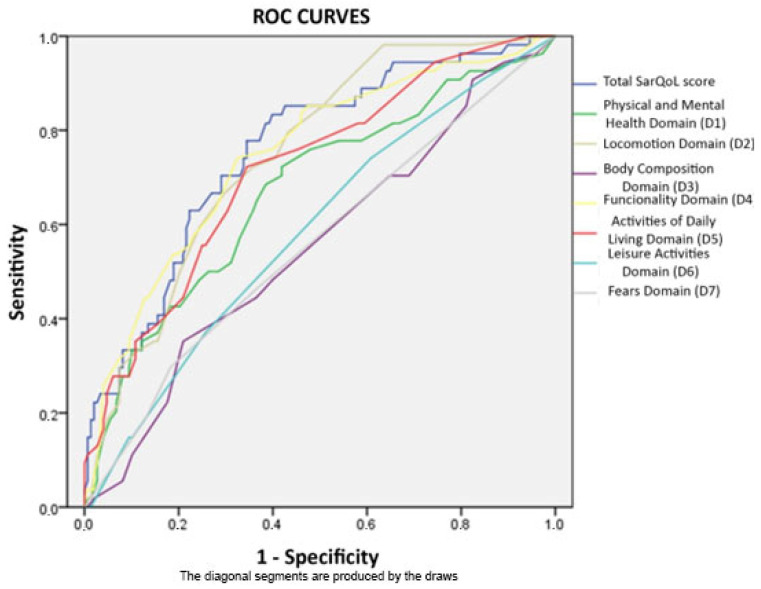
Multiple ROC curves developed for every SarQoL domain and total score.

**Figure 2 ijerph-19-08473-f002:**
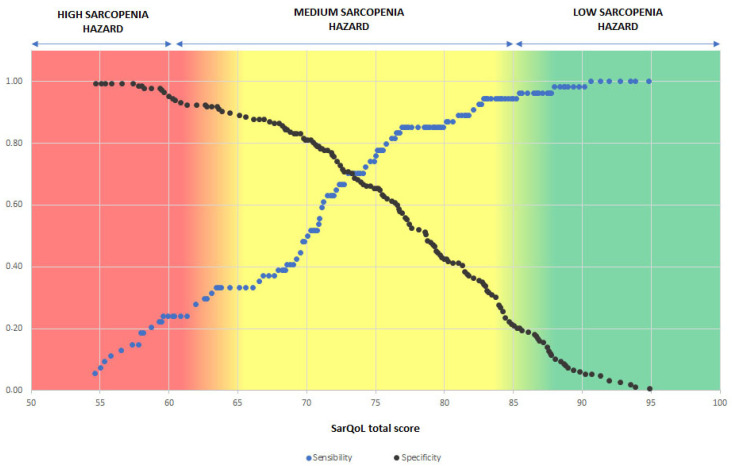
Sarcopenia hazard based on the result obtained in the SarQoL questionnaire, assessing the sensitivity and specificity for the scores obtained.

**Table 1 ijerph-19-08473-t001:** Characteristics of the study sample based on the presence of sarcopenia and the QoL.

	Total	Men		*p*-Value	Women		*p*-Value
		WS * (*N* = 27)	S ** (*N* = 11)		WS * (*N* = 121)	S ** (*N* = 43)	
Age (years)	73 ± 5	71 ± 4	75 ± 5	**0.013**	72 ± 5	74 ± 5	0.104
** *Living arrangement* **				0.230			0.062
With partner	143 (70.8%)	27 (100%)	9 (81.8%)		86 (71.1%)	21 (48.8%)	
With son	5 (2.5%)	0 (0%)	0 (0%)		3 (2.5%)	2 (4.7%)	
Alone	6 (3%)	0 (0%)	2 (18.2%)		3 (2.5%)	1 (2.3%)	
Other	48 (23.7%)	0 (0%)	0 (0%)	0.165	29 (24%)	19 (44.2%)	**0.045**
** *Education* **							
Elementary school or no degree	170 (84.1%)	19 (70.4%)	11 (100%)		100 (82.6%)	40 (94%)	
Secondary school	22 (10.9%)	8 (29.6%)	0 (0%)		12 (9.9%)	2 (4.7%)	
University or higher degree	10 (5%)	0 (0%)	0 (0%)	**0.015**	9 (7.4%)	1 (2.3%)	**0.001**
** *Frailty status (score)* **							
Non-frail	85 (42.1%)	18 (66.7%)	4 (36.4%)		54 (44.6%)	9 (20.9%)	
Pre-frail	87 (43.1%)	9 (33.3%)	4 (36.4%)		54 (44.6%)	20 (46.5%)	
Frail	30 (14.8%)	0 (0%)	3 (37.3%)	**0.017**	13 (10.7%)	14 (32.6%)	0.080
** *Nutritional status (score)* **							
Normal nourished	127 (62.9%)	22 (81.5%)	4 (36.4%)		76 (62.8%)	25 (58.1%)	
At risk of malnutrition	4 (2%)	0 (0%)	0 (0%)		1 (0.8%)	3 (7%)	
Malnourished	71 (35.1%)	5 (18.5%)	7 (63.6%)		44 (36.4%)	15 (14.9%)	
Body mass index (kg/m^2^)	27.5 ± 4.2	28.2 ± 4.4	28.7 ± 4.9	0.800	27.2 ± 4.2	27.7 ± 4.1	0.500
** *Comorbidities* **							
Hypertension	102 (50.5%)	18 (66.7%)	8 (72.7%)	0.720	54 (44.6%)	22 (51.2%)	0.460
Dyslipidemia	90 (44.6%)	14 (51.9%)	5 (45.5%)	0.620	51 (42.1%)	20 (46.5%)	0.720
Diabetes mellitus 2	28 (13.9%)	6 (22.2%)	5 (45.5%)	0.150	12 (9.9%)	5 (11.6%)	0.750
Osteoporosis	13 (6.4%)	1 (3.7%)	0 (0%)	0.520	9 (7.4%)	3 (7%)	0.920
** *SarQoL* **							
Overall quality of life	75.3 ± 10.1	81.2 ± 10.4	69.8 ± 12.5	**0.006**	76.9 ± 8.4	68.3 ± 9.6	**0.000**
Physical and mental health	75.7 ± 13.9	83.1 ± 12.7	69.3 ± 15.3	**0.007**	76.8 ± 12.7	69.6 ± 14.8	**0.003**
Locomotion	80.6 ± 15.7	86.5 ± 16.3	73 ± 15.9	**0.025**	83.6 ± 14.4	70.5 ± 13.7	**0.000**
Body composition	65.7 ± 14.9	76.2 ± 14.2	65.9 ± 15.7	0.076	64.2 ± 14.5	63.3 ± 14	0.710
Functionality	75.3 ± 13.4	80.3 ± 14.8	67.1 ± 15.3	**0.018**	77.9 ± 11.9	67.1 ± 11.7	**0.000**
Activities of daily living	72.4 ± 10.3	78.1 ± 10.3	65.7 ± 14.8	**0.005**	73.8 ± 8.3	66.6 ± 10.8	**0.000**
Leisure activities	67.9 ± 20	73.9 ± 19.2	60.5 ± 15.3	**0.046**	68.4 ± 20.3	64.6 ± 20.3	0.290
Fears	88.9 ± 17.5	94 ± 14	95.5 ± 10.1	0.750	89 ± 17.3	83.4 ± 20.2	0.110
** *Physical activity parameters* **							
METS (score)	855.4 ± 420.3	1212.1 ± 644.6	821 ± 448.1	**0.043**	845 ± 342.9	669.5 ± 294.2	**0.003**
ASMM (kg)	17.8 ± 3.6	23.2 ± 2.9	23.1 ± 3.2	0.990	16.5 ± 2.4	16.6 ± 2.8	0.850
Handgrip strength (kg)	22.4 ± 7.7	35.6 ± 7	30.1 ± 5	**0.023**	21.4 ± 3.8	15.1 ± 4.3	**0.000**
Gait speed (m/s)	1.01 ± 0.22	1 ± 0.2	0.9 ± 0.2	**0.043**	1.1 ± 0.2	0.9 ± 0.2	**0.000**
Sit to stand test (s)	11.9 ± 3.6	10.8 ± 2.6	17.4 ± 3.4	**0.000**	10.5 ± 2.1	14.9 ± 4.4	**0.000**

* WS: Without sarcopenia; S **: Sarcopenia cases; Group differences: Chi square test or Fisher’s exact test for categorical data. *t* test or Mann–Whitney U test for continuous data; The data are presented in mean ± standard deviation or *N* (percentages); *p*-Value ≤ 0.05 shown in bold.

**Table 2 ijerph-19-08473-t002:** Correlations between principal variables related with Sarcopenia and SarQoL Domains.

		Age	BMI	Weekly Physical Activity (METS)	Appendicular Skeletal Muscle Mass	Handgrip Strength	Sit to Stand Test	Gait Speed
Physical and mental health (D1)	Men	**0.341**	−0.291	**0.422**	−0.13	0.133	**−0.565**	**0.363**
	Women	**−0.202**	**−0.164**	**0.331**	−0.077	0.147	**−0.240**	0.096
Locomotion (D2)	Men	**−0.351**	**−0.412**	**0.538**	−0.24	0.198	**−0.575**	**0.544**
	Women	**−0.247**	**−0.284**	**0.424**	−0.106	**0.328**	**−0.381**	**0.351**
Body composition (D3)	Men	−0.062	−0.265	0.132	−0.088	−0.04	−0.176	−0.022
	Women	**−0.235**	0.123	**0.195**	0.079	−0.059	−0.039	0.104
Functionality (D4)	Men	**−0.390**	**−0.344**	**0.510**	−0.161	**0.397**	**−0.589**	**0.514**
	Women	**−0.198**	**−0.301**	**0.267**	−0.153	**0.267**	**−0.396**	**0.384**
Activities of daily living (D5)	Men	**−0.361**	**−0.365**	**0.331**	−0.224	0.269	**−0.422**	**0.413**
	Women	**−0.236**	**−0.168**	**−0.316**	−0.038	**0.326**	**−0.201**	**−0.281**
Leisure activities (D6)	Men	−0.232	−0.16	**0.387**	−0.074	0.251	−0.262	0.097
	Women	**−0.243**	0.011	**0.332**	0.015	0.061	**−0.231**	**0.163**
Fears (D7)	Men	−0.068	**−0.368**	0.268	−0.051	0.171	−0.094	0.252
	Women	−0.106	**−0.238**	0.016	−0.13	0.072	**−0.192**	**0.255**
SarQoL Total	Men	**−0.350**	**−0.423**	**0.521**	−0.21	**0.33**	**−0.592**	**0.513**
	Women	**−0.305**	**−0.279**	**0.406**	−0.120	**0.314**	**−0.394**	**0.383**

Pearson’s correlation coefficient for normally distributed data and Spearman’s correlation coefficient for data that are not normally distributed; *p* ≤ 0.05 results area shown in bold.

**Table 3 ijerph-19-08473-t003:** Multiple Linear Regression Analysis with Total and Domain Scores of SarQol as Dependent Variable.

Independent Variable	Men								Independent Variable	Women							
	B	SE	Beta	T	*p*	R^2^	Adjusted R^2^	Model Significance		B	SE	Beta	T	*p*	R^2^	Adjusted R^2^	Model Significance
*D1 Physical and mental health*									*D1 Physical and mental health*								
1. Age	−0.323	0.509	−0.999	−0.364	0.53	0.44	0.37	**0.001**	1. Age	−0.306	0.210	−0.116	−1.455	0.148	0.152	0.13	**0.0001**
2. METS	0.005	0.003	0.22	1.531	0.135				2. BMI	−0.441	0.245	−0.135	−1.802	0.074			
3. Sit To Stand Test	−1.655	0.609	−0.469	−2.715	0.01				3. METS	0.009	0.003	0.235	3.007	0.003			
4. Gait Speed	2.503	12.22	0.034	0.205	0.839				4. Sit To Stand Test	−0.595	0.302	−0.151	−1.970	0.051			
*D2 Locomotion*									*D2 Locomotion*								
1. Age	−0.313	0.498	−0.082	−0.628	0.534	0.628	0.57	**0.0001**	1. Age	−0.460	0.208	−0.154	−2.215	0.028	0.339	0.323	**0.0001**
2. BMI	−1.568	0.424	−0.411	−3.703	0.001				2. BMI	−1.296	0.241	−0.352	−5.368	0.000			
3. METS	0.007	0.003	0.254	2.128	0.041				3. METS	0.013	0.003	0.287	4.178	0.000			
4. Sit To Stand Test	−0.634	0.598	−0.154	−0.106	0.297				4. Handgrip	0.760	0.212	0.238	3.581	0.000			
5. Gait Speed	27.324	11.810	0.320	2.314	0.027												
*D3 Body Composition*									*D3 Body Composition*								
									1. Age	−0.583	0.224	−0.209	−2.604	0.010	0.066	0.054	**0.004**
									2. METS	0.004	0.003	0.097	1.209	0.228			
*D4 Functionality*									*D4 Functionality*								
1. Age	−0.134	0.527	−0.038	−0.254	0.801	0.6	0.522	**0.0001**	1. Age	−0.258	0.191	−0.104	−1.353	0.178	0.312	0.285	**0.0001**
2. BMI	−1.170	0.427	−0.330	−2.736	0.010				2. BMI	−0.898	0.213	−0.293	−4.214	0.000			
3. METS	0.006	0.003	0.239	1.894	0.068				3. METS	0.004	0.003	0.116	1.632	0.105			
4. Handgrip	0.502	0.323	0.219	1.556	0.130				4. Handgrip	0.286	0.188	0.107	1.515	0.132			
5. Sit To Stand Test	−0.977	0.587	−0.256	−1.667	0.106				5. Sit To Stand Test	−0.916	0.287	−0.247	−3.192	0.002			
6. Gait Speed	15.118	11.875	0.191	1.273	0.212				6. Gait Speed	6.286	4.680	0.112	1.343	0.181			
*D5 Activities of Daily Living*									*D5 Activities of Daily Living*								
1. Age	−0.586	0.447	−0.206	−1.310	0.199	0.47	0.387	**0.001**	1. Age	−0.219	0.153	−0.118	−1.433	0.154	0.211	0.181	**0.0001**
2. BMI	−0.799	0.381	−0.278	−2.098	0.044				2. BMI	−0.388	0.171	−0.169	−2.267	0.025			
3. METS	0.003	0.003	0.146	1.023	0.314				3. METS	0.005	0.002	0.159	2.086	0.039			
4. Sit To Stand Test	−0.185	0.538	−0.060	−0.344	0.733				4. Handgrip	0.434	0.151	0.217	2.868	0.005			
5. Gait Speed	22.902	10.618	0.356	2.157	0.039				5. Sit To Stand Test	−0.262	0.230	−0.094	−1.136	0.258			
									6. Gait Speed	2.985	3.754	0.071	0.795	0.428			
*D6 Leisure Activities*									*D6 Leisure Activities*								
1. METS	0.008	0.005	0.267	1.665	0.105	0.071	0.046	**0.105**	1. Age	−0.519	0.328	−0.132	−1.582	0.116	0.113	0.103	**0.000**
									2. METS	0.015	0.005	0.243	3.073	0.002			
									3. Sit To Stand Test	−0.580	0.508	−0.099	−1.141	0.256			
									4. Gait Speed	0.509	8.009	0.006	0.063	0.949			
*D7 Fears*									*D7 Fears*								
1. BMI	−1.337	0.424	−0.465	−3.155	0.003	0.217	0.195	**0.003**	1. BMI	−0.761	0.329	−0.174	−2.311	0.022	0.117	0.101	**0.0001**
									2. Sit To Stand Test	−0.591	0.452	−0.112	−1.307	0.193			
									3. Gait Speed	1.573	6.890	0.197	2.282	0.024			
*SarQol Total*									*SarQol Total*								
1. Age	−0.340	0.351	−0.127	−0.968	0.341	0.693	0.634	**0.0001**	1. Age	−0.289	0.135	−0.157	−2.137	0.034	0.375	0.351	**0.0001**
2. BMI	−1.115	0.285	−0.414	−3.918	0.000				2. BMI	−0.605	0.151	−0.265	−3.996	0.000			
3. METS	0.005	0.002	0.270	2.451	0.020				3. METS	0.007	0.002	0.234	3.439	0.001			
4. Handgrip	0.251	0.215	0.144	1.169	0.251				4. Handgrip	0.299	0.134	0.151	2.239	0.027			
5. Sit To Stand Test	−0.607	0.390	−0.209	−1.556	0.130				5. Sit To Stand Test	−0.608	0.204	−0.221	−2.985	0.003			
6. Gait Speed	12.555	7.903	0.208	1.589	0.122				6. Gait Speed	3.364	3.321	0.081	1.013	0.313			

*p* ≤ 0.05 results area shown in bold.

**Table 4 ijerph-19-08473-t004:** ROC curves diagnostic performance values for each domain and total score of SarQoL questionnaire when it was used to discriminate sarcopenic pathology.

Analyzed Variables	95% Asymptotic Confidence Interval			
	Area	*p*-Value	Lower Limit	Upper Limit
Total SarQoL score	**0.756 ***	0.000	0.682	0.83
Physical and Mental Health Domain (D1)	0.671	0.000	0.584	0.758
Locomotion Domain (D2)	**0.749 ***	0.000	0.679	0.819
Body Composition Domain (D3)	0.554	0.244	0.464	0.644
Functionality Domain (D4)	**0.746 ***	0.000	0.669	0.824
Activities Of Daily Living Domain (D5)	**0.717 ***	0.000	0.638	0.796
Leisure Activities Domain (D6)	0.586	0.063	0.498	0.673
Fears Domain (D7)	0.559	0.202	0.467	0.65

* Area >0.70 is shown in bold.

**Table 5 ijerph-19-08473-t005:** ROC curves diagnostic performance values for each domain and total score of SarQoL questionnaire when it was used to discriminate sarcopenic pathology (only confirmed and severe cases).

Analyzed Variables	95% Asymptotic Confidence Interval			
	Area	*p*-Value	Lower Limit	Upper Limit
Total SarQoL score	0.685	0.014	0.556	0.815
Physical and Mental Health Domain (D1)	0.647	0.052	0.506	0.787
Locomotion Domain (D2)	0.696	0.009	0.576	0.817
Body Composition Domain (D3)	0.455	0.553	0.302	0.608
Functionality Domain (D4)	0.621	0.109	0.489	0.753
Activities Of Daily Living Domain (D5)	0.666	0.028	0.533	0.799
Leisure Activities Domain (D6)	0.661	0.032	0.530	0.793
Fears Domain (D7)	0.515	0.838	0.372	0.659

## Data Availability

The data presented in this study are available on request from the corresponding author.

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
