# Peer review of "SarQoL Questionnaire in Community-Dwelling Older Adults under EWGSOP2 Sarcopenia Diagnosis Algorithm: A New Screening Method?"

_ijerph, 2022, doi:10.3390/ijerph19148473_

Round 1
Reviewer 1 Report
This is a very interesting manuscript highlighting a probable new tool for the screening of sarcopenia.
I have few comments to address to authors:
1/ In your paper, the prevalence of sarcopenia is very high. This is probably due to the fact that authors combined probable, confirmed and severe sarcopenia to define a sarcopenic status. The consideration of patients with probable sarcopenia in the sarcopenic group is probably something that needs to be discussed for this paper. I would suggest to authors to run extra analyses to confirm the results when restricting the sarcopenic population to those with a confirmed diagnosis of sarcopenia using the EWGSOP2 criteria. Those extra analyses could be presented in appendix.
2/ I think authors missed an important study to compare their results with (https://pubmed.ncbi.nlm.nih.gov/34212342/). It is mandatory that authors took this paper into account in their discussion section.
3/ Authors mentioned the use of IPAQ-E in the “sarcopenia diagnosis” section of their manuscript. However, this questionnaire is not relevant for the diagnosis and should be moved into another section.
4/ Authors should replace the commas by dots in all the tables.
Author Response
Reviewer 1
Reviewer’s comment:
“In your paper, the prevalence of sarcopenia is very high. This is probably due to the fact that authors combined probable, confirmed and severe sarcopenia to define a sarcopenic status. The consideration of patients with probable sarcopenia in the sarcopenic group is probably something that needs to be discussed for this paper. I would suggest to authors to run extra analyses to confirm the results when restricting the sarcopenic population to those with a confirmed diagnosis of sarcopenia using the EWGSOP2 criteria. Those extra analyses could be presented in appendix.”
Author’s comment:
According to reviewer’s comment, an extra analysis of the data has been carried out including only the population with confirmed and severe sarcopenia. The new table has been added at the results. The results obtained from these calculations have decreased the total discriminative capacity of the questionnaire and that of its domains independently.
These changes could be explained by the introduction of spectrum BIAS in diagnostic studies.
The prevalence of cases is lower a part of cases are considered now as controls (non-cases). This could lead to an over or even an infrastimation of diagnostic performance of test.
Added a new paragraph in discussion section to clarify the inclusion of patients with probable sarcopenia within the sarcopenic group.
Reviewer’s comment:
“I think authors missed an important study to compare their results with (https://pubmed.ncbi.nlm.nih.gov/34212342/). It is mandatory that authors took this paper into account in their discussion section.”
Author’s comment:
Thank you for your advice. We have added a full paragraph in the discussion section referencing this article and showing the main common and disagreeing points.
Reviewer’s comment:
“Authors mentioned the use of IPAQ-E in the “sarcopenia diagnosis” section of their manuscript. However, this questionnaire is not relevant for the diagnosis and should be moved into another section.”
Author’s comment:
According to reviewer’s comment, IPAQ-E item has been removed from “Physical Activity Performance” section on “sarcopenia diagnosis”. It has been added as an additional item on “Materials and Methods” index.
Reviewer’s comment:
“Authors should replace the commas by dots in all the tables.”
Author’s comment:
According to reviewer’s input, we replace commas by dots in all the tables.

Reviewer 2 Report
Thank you for the opportunity of reviewing the valuable manuscript.
Since sarcopenia probable is also included and judged as a sarcopenia group, we believe it is reasonable that locomotion is relevant in SarQol, the result of this study. However, I believe that such a system, which can be easily checked as a screening tool, would be useful for early detection. The sensitivity of 0.749 is also excellent. However, since the number of participants was relatively small (about 200), we would like you to mention the possibility of bias in the results and the presence of some false-negative results, and please describe them in the limits. Furthermore, we believe that the paper would be useful to beginners in preventive medicine if you could include examples of how the authors think it could be used in specific situations.
In addition, the following minor corrections are also requested.
・The subheadings in Table 1 are difficult to read. Please organize them.
・The font is different in some parts of the text. Please correct.
Best regards.
Author Response
Journal: IJERPH (ISSN 1660-4601)
Manuscript ID: ijerph-1786396
Title
SarQol Questionnaire in Community-Dwelling Older Adults Under EWGSOP2 Sarcopenia Diagnosis Algorithm: A New Screening Method?
Carlos Guillamon-Escudero, Angela Diago-Galmés, David Zuazua Rico, Alba Maestro-González, Jose M. Tenías-Burillo, Jose M. Soriano and Julio J. Fernández-Garrido
Reviewer 2
Reviewer’s comment:
“Thank you for the opportunity of reviewing the valuable manuscript.
Since sarcopenia probable is also included and judged as a sarcopenia group, we believe it is reasonable that locomotion is relevant in SarQol, the result of this study. However, I believe that such a system, which can be easily checked as a screening tool, would be useful for early detection. The sensitivity of 0.749 is also excellent. However, since the number of participants was relatively small (about 200), we would like you to mention the possibility of bias in the results and the presence of some false-negative results, and please describe them in the limits. Furthermore, we believe that the paper would be useful to beginners in preventive medicine if you could include examples of how the authors think it could be used in specific situations.”
Author’s comment:
The reviewer's appreciation according to the possibility of bias and false negatives was added in the discussion section.
The authors added a paragraph in the discussion section referring to the possible example of this screening model (line 341).
Reviewer’s comment:
“The subheadings in Table 1 are difficult to read. Please organize them.”
Author’s comment:
According to the reviewer’s comment, authors have changed the subheadings format, adding bold and separating each of the subheadings to appreciate them better.
Reviewer’s comment:
“The font is different in some parts of the text. Please correct.”
Author’s comment:
According to the reviewer’s input, it has been checked and changed in all parts of the text.
